# Reinforced Data-Driven Estimation for Spectral Properties of Koopman Semigroup in Stochastic Dynamical Systems

## Abstract

Analyzing the spectral properties of the Koopman operator is crucial for understanding and predicting the behavior of complex stochastic dynamical systems. However, the accuracy of data-driven estimation methods, such as Extended Dynamic Mode Decomposition (EDMD) and its variants, are heavily dependent on the quality and location of the sampled trajectory data. This paper introduces a novel framework, Reinforced Stochastic Dynamic Mode Decomposition, which integrates Reinforcement Learning (RL) with Stochastic Dynamic Mode Decomposition (SDMD) to automatically guide the data collection process in stochastic dynamical systems. We frame such strategy as an RL problem, where an agent learns a policy to select trajectory initial conditions. The agent is guided by a reward signal based on *spectral consistency*, that is a measure of how well the estimated Koopman eigenpairs describe the system's evolution balanced with an exploration bonus to ensure comprehensive coverage of the state space. We demonstrate the effectiveness of our approach using Bandit algorithm, Deep Q-Network (DQN), and Proximal Policy Optimization (PPO) algorithms on canonical systems including the double-well potential, the stochastic Duffing oscillator and the FitzHugh-Nagumo model. Our results show that the RL agent automatically discovers dynamically significant regions without any prior knowledge of the system. Rigorous theoretical analysis establishes convergence guarantees for the proposed algorithms, directly linking the final estimation accuracy to the quality of the learned policy. Our work presents a robust, automated methodology for the efficient spectral analysis of complex stochastic systems.

## 1 Introduction

The challenge of understanding, modeling, and predicting the behavior of complex dynamical systems is a central theme across science and engineering Strogatz (2001); Hirsch et al. (2013); Brunton & Kutz (2022); Hale (2009); Arnold (2013). Traditionally, this challenge has been met with first-principle models Holmes (2012); Murray (2007); Batchelor (2000). However, for many modern systems, such as turbulent fluid flow and neural systems, the underlying mechanisms are too complex to model explicitly. With the increasing availability of data, this has brought a shift towards data-driven methods, which leverage computational power to extract dynamical patterns directly from observational data Brunton et al. (2016); Kutz et al. (2016); Williams et al. (2015a); Brunton & Kutz (2022). One particularly noteworthy data-driven approach is the Koopman operator framework, which has gained significant attention Koopman (1931); Koopman & von Neumann (1932). It offers a powerful alternative by recasting nonlinear dynamics into an infinite-dimensional, but linear evolution in a space of observable functions. The spectral properties, i.e., eigenvalues and eigenfunctions, of the Koopman operator provide profound insights into the system's behavior, revealing conserved quantities, long-term trends, metastable states, and oscillatory modes Mezić (2005).

The pursuit of these spectral properties from data has led from the seminal Dynamic Mode Decomposition (DMD) Rowley et al. (2009); Tu (2013) to advanced variants such as Extended DMD (EDMD) Williams et al. (2015a), kernel-EDMD Williams et al. (2015b), Jet-EDMD Ishikawa et al. (2024), Residual DMD (ResDMD) Colbrook & Townsend (2024), and Stochastic DMD (SDMD) Xu et al. (2025a), which address challenges of nonlinearity, high-dimensional features, eigenfunction

interpretation, and stochasticity. A persistent issue in these methods is the choice of observable functions, or *dictionary* Williams et al. (2015a); Xu et al. (2025a); Klus et al. (2020); Tu (2013); Xu et al. (2025b); Ishikawa et al. (2024); Li et al. (2017); Kreider et al. (2025); Schütte et al. (2023); **?**); Hou et al. (2023); Devergne et al. (2024); Kostic et al. (2024). While hand-crafted bases may succeed for specific systems, they require expert knowledge which motivates dictionary learning Li et al. (2017); Donoho (2006) via neural networks. Yet, the quality of learned bases is highly dependent on the data, i.e., sampling from noisy or uninformative regions produces inaccurate decompositions. Thus, good data is essential but difficult to identify without prior dynamical knowledge. To address this, we propose Reinforced SDMD, which uses Reinforcement Learning (RL) Sutton & Barto (1998) to automate data acquisition. An RL agent learns policies for selecting trajectory initial points by maximizing a reward that reflects spectral accuracy that balances exploitation of informative regions with exploration of new ones. The present paper is devoted to the development of a comprehensive Reinforced SDMD framework. We introduce an RL-based framework for intelligent data-collection in Koopman analysis, guided by a reward function based on *spectral consistency*. This approach is validated with Multi-armed Bandit Slivkins (2019), DQN Mnih et al. (2013), and PPO Schulman et al. (2017) algorithms on benchmark systems, showing it can automatically identify key dynamical features. We also provide convergence analysis that links the quality of the collected data to the final estimation accuracy. This work pioneers a new paradigm that fully integrates intelligent data acquisition into the process of Koopman spectral analysis.

The paper is organized as follows. Section 2 provides the necessary background on the stochastic Koopman operator and the SDMD algorithm. Section 3 details our proposed methodology after introducing the three classical reinforced learning schemes. Section 4 presents the experimental results on several stochastic dynamical systems. Section 5 provides the theoretical convergence analysis for our methods. Finally, we conclude with a summary and discussion of future work.

## 2 PRELIMINARIES

### 2.1 STOCHASTIC KOOPMAN OPERATOR

The Koopman operator framework provides a powerful mathematical tool for analyzing dynamical systems by transforming nonlinear dynamics into a linear representation in an infinite-dimensional function space. For stochastic systems, this framework extends naturally to capture probabilistic evolution over time.

Consider a continuous-time stochastic process $(X_t)_{t \geq 0}$ on a probability space $(\Omega, \mathbb{P})$ defined by the stochastic differential equation (SDE):

$$dX_t = b(X_t) \, dt + \sigma(X_t) \, dW_t, \quad X_0 = x \in \mathcal{M}, \tag{2.1}$$

where $\mathcal{M} \subseteq \mathbb{R}^d$ is the state space, $b : \mathcal{M} \to \mathbb{R}^d$ is the drift, $\sigma : \mathcal{M} \to \mathbb{R}^{d \times m}$ is the diffusion coefficients, and $(W_t)_{t \geq 0}$ is a standard $m$-dimensional Wiener process.

Let $\rho$ be a probability measure on $\mathcal{M}$ and denote by $\mathcal{F} := L^2(\mathcal{M}, \rho)$ the space of observables. The space $\mathcal{F}$ is equipped with the usual inner product $\langle f, g \rangle_\rho := \int_\mathcal{M} fg \, d\rho$ and corresponding norm $\|f\|_\rho := \sqrt{\langle f, f \rangle_\rho}$. The stochastic Koopman semigroup $(\mathcal{K}^t)_{t \geq 0}$ on $\mathcal{F}$ is defined as: for $f \in \mathcal{F}$, $x \in \mathcal{M}$, and $t \geq 0$,

$$(\mathcal{K}^t f)(x) := \mathbb{E}_\mathbb{P}[f(X_t)|X_0 = x], \tag{2.2}$$

where $\mathbb{E}_\mathbb{P}$ denotes the expectation with respect to probability measure $\mathbb{P}$. This operator captures how the expected value of observables evolves under random perturbation.

The infinitesimal generator $\mathcal{A}$ of the Koopman semigroup is given by:

$$\mathcal{A}f := \lim_{t \to 0} \frac{\mathcal{K}^t f - f}{t}, \tag{2.3}$$

on the domain $\mathcal{D}(\mathcal{A}) = \left\{ f \in \mathcal{F} : \lim_{t \to 0} \frac{\mathcal{K}^t f - f}{t} \text{ exists in } \mathcal{F} \right\}$. By Itô's formula Pavliotis (2016),

$$\mathcal{A}f = \sum_{i=1}^d b_i \frac{\partial f}{\partial x_i} + \frac{1}{2} \sum_{i,j=1}^d (\sigma \sigma^T)_{ij} \frac{\partial^2 f}{\partial x_i \partial x_j}, \quad \forall f \in C_b^2(\mathcal{M}). \tag{2.4}$$

**Assumption 2.1.** Throughout this paper, we assume that $\{\mathcal{K}^t\}_{t\geq 0}$ is a $C_0$-semigroup on $\mathcal{F}$ Pavliotis (2016).

The Assumption 2.1 guarantees that not only is the domain $\mathcal{D}(\mathcal{A})$ dense in $\mathcal{F}$ but also the generator $\mathcal{A}$ is a closed operator in $\mathcal{F}$ Pazy (2012).

**Remark 2.2.** For each $t > 0$, the eigenvalues $\lambda$ of the Koopman generator $\mathcal{A}$ and the eigenvalues $\mu$ of the stochastic Koopman operator $\mathcal{K}^t$ can be related by

$$\mu = e^{t\lambda}. \tag{2.5}$$

## 2.2 Stochastic Dynamic Mode Decomposition (SDMD)

Traditional Extended Dynamic Mode Decomposition (EDMD) Williams et al. (2015a) was designed for estimating the Koopman operator on a finite dimensional subspace in deterministic systems and does not explicitly account for stochastic effects. SDMD Xu et al. (2025a) addressed this limitation by incorporating stochastic Taylor expansion (Pavliotis, 2016, section 5.2) into the approximation framework, providing enhanced numerical stability and precision for stochastic systems.

The key innovation of SDMD lies in its explicit incorporation of sampling time $\Delta t$ via *stochastic Taylor expansion* in the approximation process. Given a dictionary of basis functions $\{\psi_1, \ldots, \psi_N\} \subset \mathcal{D}(\mathcal{A})$ forming the finite-dimensional space $\mathcal{F}_N := \text{span}\{\psi_1, \ldots, \psi_N\}$, SDMD constructs empirical Gram matrices from i.i.d. data $\{x_k\}_{k=1}^m$:

$$\widehat{G} = \frac{1}{m}\Psi_X^*\Psi_X, \quad \widehat{H} = \frac{1}{m}\Psi_X^*\Psi_X', \tag{2.6}$$

where $[\Psi_X]_{ij} := \psi_j(x_i)$ contains evaluation of basis functions and $[\Psi_X']_{ij} := \mathcal{A}\psi_j(x_i) = \frac{d}{dt}\psi_j(x_i)$ contains evaluations of the time differential of basis functions Klus et al. (2020). Note that $*$ denotes the conjugate transpose of a matrix or adjoint operator. The SDMD approximation of the Koopman operator is then computed as following:

$$\widehat{K}_{N,\Delta t,m} = I + \Delta t\,\widehat{G}^{-1}\widehat{H}. \tag{2.7}$$

This formulation directly approximates the Koopman semigroup rather than its generator, avoiding computationally expensive matrix exponential calculations while ensuring numerical stability. The method provides rigorous convergence guarantees across three critical limits: large data ($m \to \infty$), infinitesimal sampling time ($\Delta t \to 0$), and increasing dictionary size ($N \to \infty$) (See (Xu et al., 2025a, Section 4) for more details).

In order to train the dictionary directly from data without manual intervention, SDMD can also be extended with dictionary learning Li et al. (2017); Donoho (2006), namely SDMD-DL, where basis functions $\Psi(x; \theta)$ are parameterized by neural networks parameters $\theta$ and optimized via the following minimization problem:

$$J(\theta) = \|\Psi_Y(\theta) - \Psi_X(\theta)\widehat{K}_{N,\Delta t,m}(\theta)\|_F^2 + \gamma\mathcal{R}_{\widehat{K}}, \tag{2.8}$$

where $\mathcal{R}_{\widehat{K}}$ is the regularization term.

## 3 Methodology: Reinforced Koopman Operator Learning

Reinforcement learning provides a framework for sequential decision-making where an agent learns optimal policy through interaction with an environment. In our context, RL algorithms guide the selection of optimal initial conditions for trajectory generation to improve Koopman operator approximation quality. We consider three representative RL approaches with varying capabilities: Bandit with $\epsilon$-greedy Slivkins (2019), DQN Mnih et al. (2013) and PPO Schulman et al. (2017). In this section, we first briefly introduce these three methods, and then we discuss how to incorporate them with SDMD method to learn Koopman operator.

### 3.1 Review of RL Algorithms

We briefly review three reinforcement learning algorithms adapted in our framework, each representing a different approach to decision-making.

**Multi-armed Bandit with $\varepsilon$-greedy Policy:** The multi-armed bandit problem Slivkins (2019) is the simplest, stateless form of RL. Each action $a$ corresponds to selecting a spatial region for trajectory initialization. The agent maintains an estimated reward $Q(a)$, updated after each trial. Using an $\varepsilon$-greedy policy, it balances exploitation of the best-known action with random exploration, providing a simple yet effective baseline for sampling.

**Deep Q-Network (DQN):** DQN Mnih et al. (2013) extends the bandit by incorporating *state*, here represented by a history of sampling locations. A neural network approximates the Q-function $Q(S_t, a_t; \theta)$ and is trained to satisfy temporal consistency through TD updates. Practical techniques such as experience replay and a target network stabilize training. Unlike the bandit's fixed exploration rate, DQN typically uses a decaying $\varepsilon$-greedy policy.

**Proximal Policy Optimization (PPO):** PPO Schulman et al. (2017) is a policy-based method that directly learns $\pi_\theta(a|s)$ using an actor-critic architecture. The actor selects actions while the critic estimates value functions. Policy updates are constrained by a clipped surrogate objective to avoid instability, with advantages estimated via GAE. PPO alternates between data collection and policy updates, which makes it effective for complex control tasks.

**Remark 3.1.** See Appendix A.1 for more detailed review.

### 3.2 Reinforced SDMD

The core idea of our method is to guide an RL agent's data collection strategy using a reward signal derived from Koopman spectral analysis. We define three key components that form the interface between RL and SDMD.

**Action Space.** An action $a_t$ corresponds to selecting a predefined region (a "box" in a discretized domain) from which to initialize a new data trajectory.

**State Representation.** For state-dependent algorithms like DQN and PPO, the state $S_t \in \mathbb{R}^{d \times \ell}$ is a sliding window containing the last $\ell$ trajectory starting points. This provides the agent with a memory of its recent exploration history.

**Reward Formulation.** The immediate reward $R_t$ balances exploitation and exploration. The exploitation term is driven by *spectral consistency*, which measures how well the SDMD-estimated eigenpairs $(\mu_i, \phi_i)$ predict the system's evolution on a given dataset $(X, Y)$:

$$\mathcal{L}_{Spectral\ Consistency} := \sum_i^N \|\phi_i(Y) - \mu_i \phi_i(X)\|_\rho^2. \tag{3.1}$$

A lower spectral consistency error implies a better model and thus a higher reward. The exploration term encourages visiting less-sampled regions, implemented via a Gaussian kernel density estimate $\eta(x_{\text{new}})$ over past starting points. The combined reward is:

$$R_t := R_0 - \mathcal{L}_{Spectral\ Consistency} + \alpha_{exp} \cdot \frac{1}{\eta(x_{\text{new}}) + \varepsilon}, \tag{3.2}$$

where $R_0$ is a baseline constant and $\alpha_{exp}$ is exploration coefficient. With these components defined, we now specify how each RL algorithm is integrated.

**Bandit-SDMD** The bandit approach is stateless and treats each grid cell as an independent arm. It learns a Q-value $Q(a)$ for each action by incrementally averaging the rewards obtained from that region. After convergence, the $Q$-values form a reward map that highlights dynamically significant regions-such as those near equilibria or invariant manifolds without requiring any prior system knowledge. The pseudocode is given in Algorithm 1.

**DQN-SDMD** The DQN approach utilizes the state representation $S_t$ to learn a state-dependent Q-function $Q(S_t, a_t; \theta)$ via a neural network. When an action $a_t$ is taken, a new trajectory from $x_{\text{new}}$ is generated. This new data is combined with data from the previous $\ell$ starting points in $S_t$ to form a comprehensive dataset. SDMD processes this dataset to compute the spectral consistency and, subsequently, the reward $R_t$. The network parameters $\theta$ are then updated by minimizing the Huber loss Huber (1992) over the TD error:

$$\mathcal{L}^H(S_t, a_t, R_t, S_{t+1}; \theta, \theta^-) := \text{Huber}\left(R_t + \gamma \max_{a'} Q(S_{t+1}, a'; \theta^-) - Q(S_t, a_t; \theta)\right).$$

This off-policy method Sutton & Barto (1998), combined with experience replay and a decaying exploration schedule, allows the agent to efficiently learn from past evaluations and adapt its sampling strategy. See Appendix A.3 for more details on the difference of Huber loss and standard mean squared error (MSE). The pseudocode is given in Algorithm 2.

**PPO-SDMD** The PPO approach directly optimizes a policy network $\pi(a|s; \theta_{\text{actor}})$ within an actor-critic architecture. It uses the same state representation and reward computation mechanism as DQN-SDMD. However, its learning mechanism is distinct. The policy parameters $\theta_{\text{actor}}$ are updated by maximizing a clipped surrogate objective, which ensures stable policy improvements:

$$\mathcal{L}_{\text{PPO}}^{\text{CLIP}}(\theta_{\text{actor}}) := \widehat{\mathbb{E}}_t \left[ \min \left( r_t(\theta_{\text{actor}}) \widehat{A}_t, \text{clip}\left(r_t(\theta_{\text{actor}}), 1 - \epsilon, 1 + \epsilon\right) \widehat{A}_t \right) \right]. \tag{3.3}$$

As an on-policy algorithm, PPO collects batches of data under the current policy before performing updates. This, combined with the conservative update rule, makes it particularly robust for learning in our setting, where the SDMD-derived reward landscape can be complex and non-stationary.

---

**Algorithm 1** Multi-Armed Bandit with SDMD

**Input:** Grid size $k$, steps $T_{\max}$, exploration rate $\varepsilon$, initial Q-value $Q_{\text{init}}$
1: $Q(a) \leftarrow Q_{\text{init}}, \ N(a) \leftarrow 0 \ \forall a$
2: **for** $t = 1$ to $T_{\max}$ **do**
3:      Select action $a_t$ via $\varepsilon$-greedy
4:      Execute $a_t$, observe state $(X, Y)$
5:      Compute reward $R_t$ (Eq. 3.2)
6:      Increment count $N(a_t)$
7:      Update $Q(a_t)$ with sample average
8: **end for**

---

**Algorithm 2** DQN with SDMD

**Input:** State window length $\ell$, replay capacity $N_{\text{replay}}$, parameters $(\gamma, \tau, \varepsilon\text{-schedule})$
1: Initialize policy network $Q(s, a; \theta)$ and target network $Q(s, a; \theta^-)$
2: Initialize replay buffer $\mathcal{D}$ with capacity $N_{\text{replay}}$
3: **for** $t = 1$ to $T_{\max}$ **do**
4:      Choose $a_t$ via $\varepsilon$-greedy on $Q(S_t, \cdot; \theta)$
5:      Execute $a_t$, observe reward $R_t$ and next state $S_{t+1}$
6:      Store transition in $\mathcal{D}$
7:      Sample minibatch, compute targets and perform gradient step
8:      Update target network by soft update
9: **end for**

---

**Algorithm 3** PPO with SDMD

**Input:** State window length $\ell$, batch size $N_{\text{batch}}$, parameters $(\gamma, \lambda, \epsilon, K)$
1: Initialize actor network $\pi(a|s; \theta_{\text{actor}})$ and critic network $V(s; \theta_{\text{critic}})$
2: **for** iteration $= 1, \ldots, M$ **do**
3:      Collect a batch of transitions into buffer $\mathcal{B}$
4:      Compute advantages using GAE (Eq. A.4)
5:      **for** epoch $= 1, \ldots, K$ **do**
6:          Update actor parameters via clipped surrogate objective
7:          Update critic parameters via value-function loss
8:      **end for**
9:      Update old policy parameters
10: **end for**

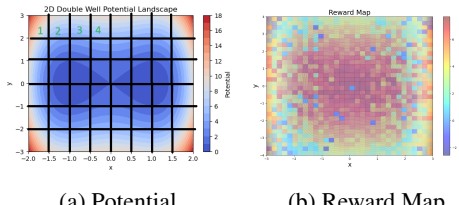

(a) Potential        (b) Reward Map

Figure 1: Potential landscape and reward map

---

**Remark 3.2.** While several recent works explore reinforcement learning or optimal design strategies for active sampling in scientific computing, these approaches focus on improving sampling efficiency rather than learning operator spectra. Our goal is fundamentally different: we use RL as a mechanism to learn the Koopman spectrum itself. Fannjiang & Listgarten (2020); Grayeli et al. (2024); Hsu (2010); Jones et al. (2024); Shen et al. (2025); Zhao & Pillai (2024)

## 4 APPLICATION

All numerical experiments were conducted on a GPU server equipped with an NVIDIA RTX 6000 Ada Generation GPU (48 GB VRAM), supported by 16 vCPUs and 62 GB of RAM. The typical training time for each experimental setup was completed within 24 hours.

## 4.1 DOUBLE WELL POTENTIAL SYSTEM

The 2D double well potential system

$$\mathrm{d}X_t = -\nabla V(X_t)\mathrm{d}t + \sigma\mathrm{d}W_t,$$

where $V(x, y) = (x^2 - 1)^2 + y^2$ and $\sigma_1 = \sigma_2 = 1.09$, is discretized into a $32 \times 32$ grid cell over domain $[-3, 3] \times [-4, 4]$ which creates $32 \times 32 = 1024$ actions. Each action selects a grid cell for trajectory initialization, as illustrated in Figure equation 1a. The algorithm learns which regions provide the best spectral approximation through trial and error iteratively.

The reward function follows Eq. equation 3.2 combining spectral consistency and exploration terms, with parameters $\varepsilon = 0.35$ for the $\varepsilon$-greedy policy and exploration coefficient $\alpha_{exp} = 0.15$. Each selected action generates a 1000-step trajectory processed through SDMD with a simple neural network, and $Q$-values are updated via sample averaging.

After 4,000 steps, the learned reward map reveals striking correspondence with the original potential landscape; more specifically, high-reward regions concentrate around the two potential minima including $(\pm 1, 0)$, as shown in Figure 1b. This correspondence is scientifically significant because regions near stable equilibria provide trajectory data with well-behaved dynamics that are ideal for Koopman analysis. The algorithm essentially rediscovered the potential landscape purely through this spectral approximation which validates the connection between dynamical importance and learning quality.

Next, the eigenfunctions evolution in Figure 2 shows clear progression from steps 100 to 4000. The first eigenfunction becomes constant, identifying the steady state. The second eigenfunction develops spatial partitioning with opposite signs in each well, capturing the metastable structure over time.

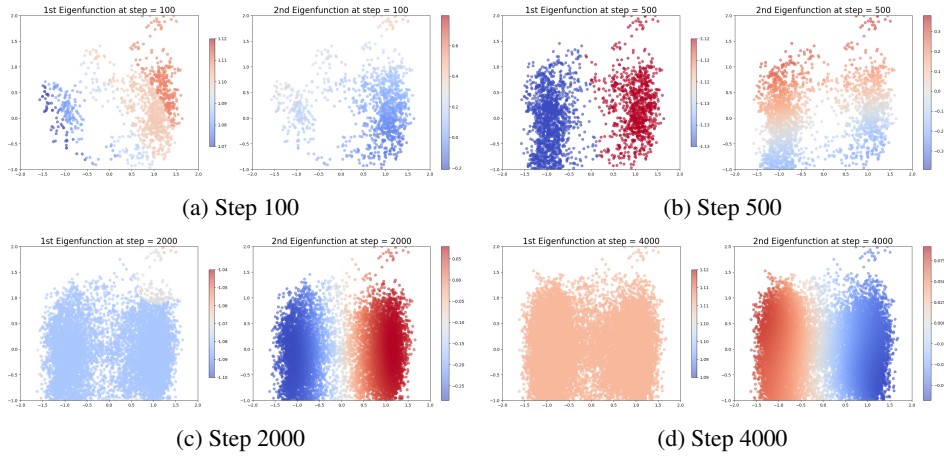

(a) Step 100         (b) Step 500

(c) Step 2000        (d) Step 4000

Figure 2: 1st & 2nd Koopman eigenfunctions of double well potential system at different time steps.

## 4.2 STOCHASTIC DUFFING OSCILLATOR

Consider the damped Duffing oscillator with noise:

$$\begin{cases} \mathrm{d}x(t) = v(t)\,\mathrm{d}t, \\ \mathrm{d}v(t) = \left(-\delta v(t) - \alpha x(t) - \beta x(t)^3\right)\mathrm{d}t + \sigma\,\mathrm{d}W(t), \end{cases} \tag{4.1}$$

with parameters $\delta = 0.5$, $\alpha = -1$, $\beta = 1$, $\sigma = 0.15$. This system has two stable equilibria at $(\pm 1, 0)$ separated by the saddle point $(0, 0)$ (Figure 3a). The damping drives trajectories into a well, while noise occasionally induces basin crossings.

Figure 3 shows the learned eigenfunctions. The first converges to a constant representing the steady state. The second captures the slowest non-trivial process, i.e., noise-induced switching between attractors. It is nearly constant within each basin but with opposite signs, which partition the phase space into regions corresponding to long-term residence before transition.

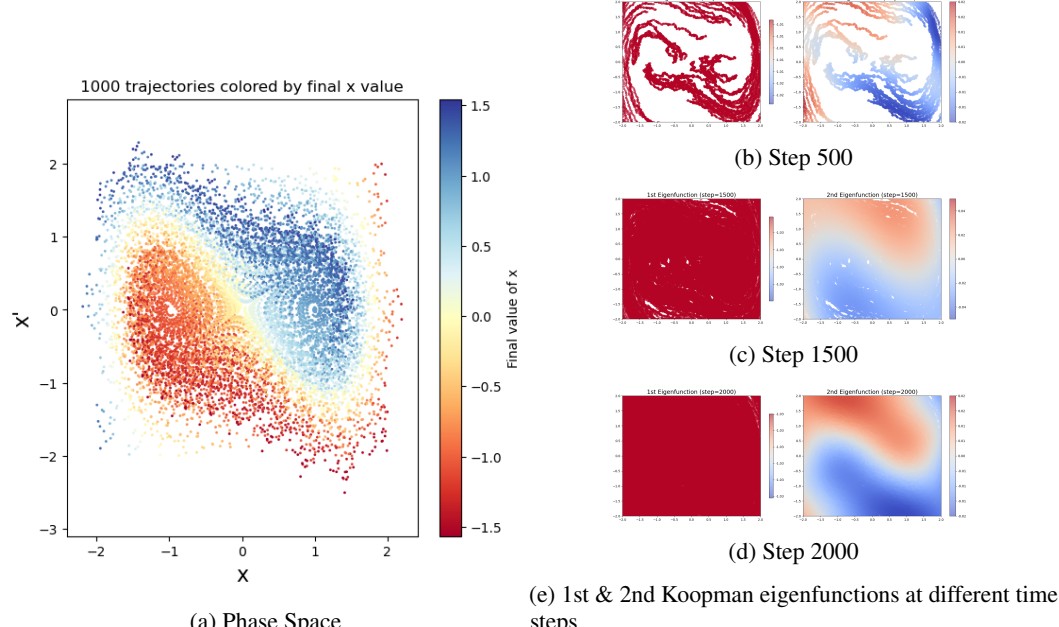

(b) Step 500

(c) Step 1500

(d) Step 2000

(a) Phase Space

(e) 1st & 2nd Koopman eigenfunctions at different time steps

Figure 3: Duffing Oscillator: Phase Space and 1st & 2nd Koopman Eigenfunctions

### 4.3 STOCHASTIC FITZHUGH-NAGUMO MODEL

We analyze the stochastic FitzHugh-Nagumo (FHN) model FitzHugh (1961); Nagumo et al. (1962), which captures excitable neuronal dynamics with a fast variable $x$ (membrane potential) and a slow variable $y$ (recovery current):

$$\begin{cases} \mathrm{d}x(t) = \big(x(t) - \frac{1}{3}x(t)^3 - y(t)\big)\,\mathrm{d}t + \sigma_1\,\mathrm{d}W_1(t), \\ \mathrm{d}y(t) = \epsilon\big(x(t) + a_1 - a_2 y(t)\big)\,\mathrm{d}t + \sigma_2\,\mathrm{d}W_2(t), \end{cases} \tag{4.2}$$

with parameters $\epsilon = 0.01$, $a_1 = 0.5$, $a_2 = 0.1$, $\sigma_1 = 10^{-3}$, $\sigma_2 = 10^{-5}$. The small $\epsilon$ creates a separation of time scales, leading to relaxation-type oscillations (Figure 4). For the deterministic case, the Koopman generator has eigenvalues $\lambda_n = -n^2\mu + in\omega$ Kato et al. (2021); Takata et al. (2023).

The dynamics alternate between slow drifts along the cubic $x$-nullcline and fast transitions near fold points ($x \approx \pm 1$). Figure 5 shows that the second eigenfunction dominates and highlights long-term attracting regions in the top-left and bottom-right of the phase space. These regions reflect slow-variable control which is distinct from potential well systems, while the middle zones correspond to rapid transitions. The unstable fixed point near $(-0.549, -0.494)$ appears as an empty area with little trajectory accumulation, and the grey strip along the nullcline separates the two dynamical phases. Figure 6 illustrates the evolution of leading Koopman eigenvalues, with further results in Appendix A.6.

## 5 CONVERGENCE ANALYSIS

In this section, we provide a theoretical analysis of the convergence properties for the Bandit-SDMD, DQN-SDMD, and PPO-SDMD algorithms. Our analysis establishes that the performance of each algorithm is fundamentally linked to the estimation precision of the SDMD Xu et al. (2025a) method.

We begin by laying out the common theoretical foundation for our analysis. Our framework operates within a standard infinite-horizon discounted Markov Decision Process (MDP), defined by the tuple $\mathcal{M} = (\mathcal{S}, \mathcal{A}, P, R, \gamma)$. Our analysis leverages a linear perspective through the Koopman operator, which acts on a dictionary of observables $\mathbf{g} : \mathcal{S} \to \mathbb{R}^N$. For a given policy $\pi$ or action $a$, the associated Koopman operator $\mathcal{K}_\pi$ or $\mathcal{K}_a$ describes the expected evolution of these observables. The cornerstone of our entire analysis is the assumption that the SDMD method provides a uniformly

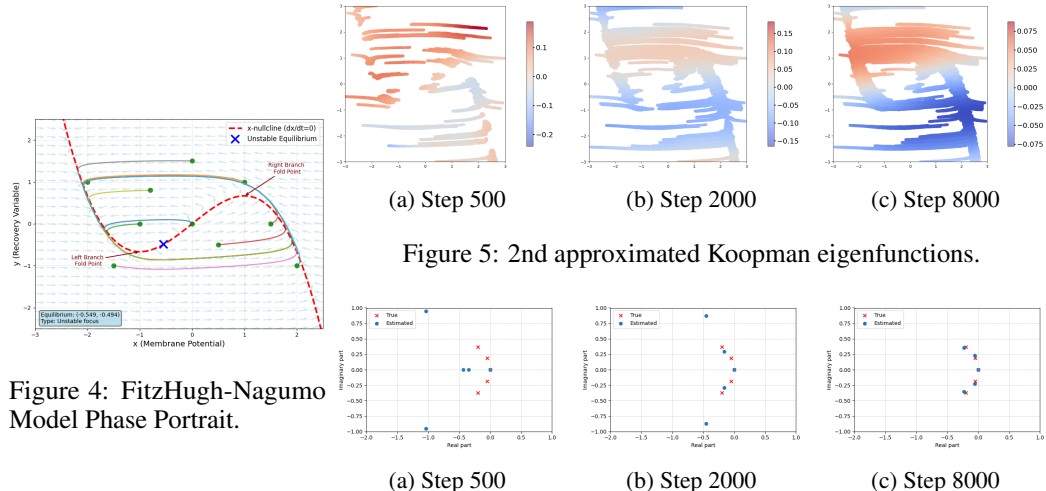

(a) Step 500       (b) Step 2000       (c) Step 8000

Figure 5: 2nd approximated Koopman eigenfunctions.

Figure 4: FitzHugh-Nagumo Model Phase Portrait.

(a) Step 500       (b) Step 2000       (c) Step 8000

Figure 6: Approximated Koopman eigenvalues.

bounded estimate $\hat{\mathcal{K}}$ of the true Koopman operator $\mathcal{K}$ on a finite dimensional space spanned by the dictionary functions.

**Assumption 5.1** (Bounded SDMD Error). Let $\mathcal{K}$ be either a policy-conditioned or action-conditioned Koopman operator, and $\hat{\mathcal{K}}$ be its estimate from SDMD. We assume the estimation error is uniformly bounded in operator norm, specifically $\|\mathcal{K} - \hat{\mathcal{K}}\| \leq \varepsilon_{\text{sdmd}}$, where the error bound $\varepsilon_{\text{sdmd}}$ is a function of dataset size with fixed sampling time and dictionary size, as established in Xu et al. (2025a).

### 5.1 REGRET ANALYSIS OF BANDIT-SDMD

For the stateless Bandit-SDMD algorithm, we analyze performance using cumulative regret, $\mathcal{R}_T = \sum_{t=1}^{T} \Delta_{a_t}$, which measures the total opportunity cost over $T$ steps. Here, each action, or arm $a$ has an unknown true mean reward $R_a$. The regret is defined by the suboptimal gap $\Delta_a := R^* - R_a$, where $R^* = \max_a R_a$ is the mean reward of the single best arm. Successful learning (i.e., achieving sub-linear regret) requires that the reward estimates from SDMD are sufficiently accurate to distinguish the optimal arm $a^*$ from any suboptimal one.

**Assumption 5.2** (Reward Estimation Precision). Let $\hat{R}_a$ be the SDMD-based estimate of the true reward $R_a$. We assume a uniform error bound $|\hat{R}_a - R_a| \leq \varepsilon_{\text{sdmd}}$.

Furthermore, we assume the error $\varepsilon_{\text{sdmd}}$ is smaller than half the minimum suboptimal gap, that is

$$\varepsilon_{\text{sdmd}} < \Delta_{\min}/2. \tag{5.1}$$

where $\Delta_{\min} := \min_{a \neq a^*} \Delta_a$.

**Theorem 5.3** (Sub-linear Regret for Bandit-SDMD). An $\epsilon$-greedy Bandit-SDMD algorithm satisfying Assumption 5.2, with a decaying exploration rate $\epsilon_t \propto 1/t$, achieves sub-linear expected cumulative regret:

$$\lim_{T \to \infty} \frac{\mathbb{E}[\mathcal{R}_T]}{T} = 0.$$

*Proof.* See Appendix B.1. □

**Remark 5.4.** If Assumption 5.2 is not satisfied (i.e., $\varepsilon_{\text{sdmd}} > \Delta_{\min}/2$), the algorithm is guaranteed to suffer linear regret, $\mathbb{E}[\mathcal{R}_T] \sim O(T)$. This is because the SDMD estimator is not precise enough to resolve the difference between the optimal arm and a close competitor, causing the agent to persistently select a suboptimal action. No exploration strategy can overcome this fundamental limitation imposed by the estimator's precision. See Appendix A.2 for more details.

## 5.2 CONVERGENCE OF DQN-SDMD

The analysis for DQN-SDMD centers on the convergence of the action-value function, $Q(s, a)$. We assume a linear representation for the Q-function, which is standard in the analysis of such algorithms.

**Assumption 5.5** (Linear Q-Function). For any stationary policy $\pi$, its action-value function $Q^\pi(s, a)$ lies in the span of a dictionary of feature functions $\mathbf{g}(s, a)$, i.e., $Q^\pi(s, a) = w_\pi^T \mathbf{g}(s, a)$ for some weight vector $w_\pi$.

Under this assumption, the error in estimating the Koopman operator $\mathcal{K}_a$ via SDMD propagates to an error in the Bellman update. See Appendix A.7 for discussion on existence of $w_\pi$. Moreover, with the help of Lemma B.1 provided in Appendix B.2, we can apply standard results from Approximate Value Iteration Munos (2005) to establish the following Theorem 5.6.

**Theorem 5.6** (Convergence of DQN-SDMD). Let $\{Q_k\}$ be the sequence of action-value functions learned by DQN-SDMD. The learned Q-function converges to a neighborhood of the optimal $Q^*$, and the value of the corresponding greedy policy $J(\pi_k)$ converges to a neighborhood of the optimal value $J(\pi^*)$. The asymptotic suboptimal gap is bounded by:

$$\limsup_{k \to \infty} \left( J(\pi^*) - J(\pi_k) \right) \leq \frac{2\gamma Q_{\max} \varepsilon_{\text{sdmd}}}{(1 - \gamma)^2}. \tag{5.2}$$

*Proof.* See Appendix B.3. □

## 5.3 CONVERGENCE OF PPO-SDMD

We model PPO-SDMD as a form of inexact Approximate Policy Iteration (API), where SDMD serves as the policy evaluation engine. We assume the state-value function $V^\pi(s)$ and the expected reward are linear in the observables $\mathbf{g}(s)$.

**Assumption 5.7** (Linear Value and Reward Function). For any stationary policy $\pi$, there exist vectors $w_\pi, r_\pi \in \mathbb{R}^d$ such that $V^\pi(s) = w_\pi^T \mathbf{g}(s)$ and $\mathbb{E}_{a \sim \pi(\cdot|s)}[R(s, a)] = r_\pi^T \mathbf{g}(s)$.

Now, the error $\varepsilon_{\text{sdmd}}$ in the SDMD estimate of $\mathcal{K}_\pi$ directly translates into an error in the policy evaluation step. The Lemma B.2 in Appendix B.4 confirms that SDMD acts as an inexact policy evaluation oracle, allowing us to bound the performance of the overall API scheme.

**Theorem 5.8** (Convergence of PPO-SDMD). The sequence of policies $\{\pi_k\}$ generated by the PPO-SDMD algorithm converges to a policy whose value $J(\pi_k)$ is near the optimal value $J(\pi^*)$. The asymptotic suboptimal gap is bounded by the SDMD estimation error as following:

$$\limsup_{k \to \infty} \left( J(\pi^*) - J(\pi_k) \right) \leq \mathcal{O}\left( \frac{\varepsilon_{\text{sdmd}}}{(1 - \gamma)^3} \right).$$

*Proof.* See Appendix B.5. □

**Remark 5.9.** The convergence results show that the performance of DQN-SDMD and PPO-SDMD is ultimately limited by the SDMD estimation error $\varepsilon_{\text{sdmd}}$. Thus, improving data quality directly enhances the estimation accuracy.

## 6 CONCLUSION

In this work, we introduced Reinforced SDMD, a framework that addresses the data-dependency challenge in Koopman spectral analysis of stochastic dynamical systems. By reframing data acquisition as a reinforcement learning problem via Koopman operator, our method is able to search for informative regions actively. With a reward signal based on spectral consistency, the agent automatically identifies key dynamical features such as potential minima and basins of attraction without prior system knowledge. This framework was validated through extensive experiments on canonical systems using Bandit, DQN, and PPO algorithms, and supported by theoretical convergence analysis. This approach establishes a new paradigm where intelligent data acquisition becomes essential to Koopman analysis. Future work includes extending to higher-dimensional spaces via continuous action representation, incorporating policy learning into dictionary optimization, and applying the method to more real-world data such as fluid dynamics and neuroscience.

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

# Appendix A

USE OF LLMS

Authors acknowledge that the use of LLMs is only for helping language polishing and grammar improvement. All other conceptual and technical parts including theoretical analysis, and experiments are made by authors.

SOURCE CODE

For reproducibility, the source code will be available at this link.

## A.1 DETAILED REVIEW OF RL ALGORITHMS

For completeness, we summarize the three reinforcement learning algorithms used in our framework with their mathematical details.

**Multi-armed Bandit with $\varepsilon$-greedy Policy.** The bandit problem Slivkins (2019) is a stateless RL setting. Each action $a$ corresponds to choosing a spatial region for trajectory initialization. The agent maintains a value estimate $Q(a)$, updated after receiving reward $R_t$:

$$Q(a_t) \leftarrow Q(a_t) + \frac{1}{N(a_t)}\big(R_t - Q(a_t)\big), \tag{A.1}$$

where $N(a_t)$ is the count of times action $a_t$ has been selected. An $\varepsilon$-greedy policy chooses the best action with probability $1 - \varepsilon$ and a random one otherwise. Despite being stateless, this method provides a strong baseline for spatial sampling.

**Deep Q-Network (DQN).** DQN Mnih et al. (2013) incorporates a notion of state $S_t$, here given by the recent history of sampling points. The action-value function $Q(S_t, a_t; \theta)$ is approximated by a neural network and trained by minimizing the temporal-difference (TD) error:

$$\delta_t = R_{t+1} + \gamma \max_{a'} Q(S_{t+1}, a'; \theta^-) - Q(S_t, a_t; \theta), \tag{A.2}$$

where $\theta^-$ is a target network. Training stability is enhanced by experience replay (sampling past transitions $(S_t, a_t, R_t, S_{t+1})$) and periodic or soft updates of $\theta^-$. Exploration follows a decaying $\varepsilon$-greedy schedule, gradually shifting from exploration to exploitation.

**Proximal Policy Optimization (PPO).** PPO Schulman et al. (2017) is a policy-based method that directly optimizes $\pi_\theta(a|s)$ using an actor-critic architecture. The actor proposes actions, and the critic estimates value $V(s)$. Its clipped surrogate objective prevents overly large policy updates:

$$\mathcal{L}^{\text{CLIP}}(\theta) = \mathbb{E}_t\Big[\min\big(r_t(\theta)\widehat{A}_t,\ \text{clip}(r_t(\theta), 1 - \varepsilon, 1 + \varepsilon)\widehat{A}_t\big)\Big], \tag{A.3}$$

where $r_t(\theta) = \pi_\theta(a_t|s_t)/\pi_{\theta_{\text{old}}}(a_t|s_t)$ and $\widehat{A}_t$ is the advantage estimate. Advantages are computed using Generalized Advantage Estimation (GAE):

$$\widehat{A}_t = \sum_{l=0}^{T-t-1} (\gamma\lambda)^l \delta_{t+l}, \quad \delta_t = R_t + \gamma V(s_{t+1}) - V(s_t). \tag{A.4}$$

PPO alternates between collecting trajectory data with the current policy and optimizing actor and critic networks, offering robustness for complex control tasks.

## A.2 ON THE FAILURE CASE OF ASSUMPTION 5.2

In this scenario, the algorithm's ability to learn is fundamentally compromised, leading to linear regret. To see why, we revisit the general regret bound established in our proof:

$$\mathbb{E}[R_T] \leq \sum_{t=1}^{T} \epsilon_t \left(\frac{1}{N}\sum_{a=1}^{N} \Delta_a\right) + \sum_{t=1}^{T}(1 - \epsilon_t)\left(\sum_{a:\Delta_a < 2\varepsilon_{\text{sdmd}}} \Delta_a\right).$$

When the condition $2\varepsilon_{\text{sdmd}} > \Delta_{\min}$ holds, the set $\{a \in \mathcal{A} \mid \Delta_a < 2\varepsilon_{\text{sdmd}}\}$ is non-empty. Let us denote the sum of gaps for these arms as $C_{\text{err}} := \sum_{a:\Delta_a < 2\varepsilon_{\text{sdmd}}} \Delta_a$. Since this set contains at least one suboptimal arm, $C_{\text{err}}$ is positive.

Consider a decaying exploration parameter $\epsilon_t = c/(Nt)$ and examine the two components of the regret bound separately. First, as shown previously, the regret from *exploration* accumulates slowly, at a logarithmic rate: $\sum_{t=1}^{T} \epsilon_t \left(\frac{1}{N} \sum_a \Delta_a\right) \sim O(\log T)$. Next, we can see that the *exploitation* term grows much faster, as illustrated in the following:

$$\sum_{t=1}^{T}(1 - \epsilon_t)C_{\text{err}} = C_{\text{err}} \sum_{t=1}^{T}(1 - \frac{c}{Nt})$$

$$= C_{\text{err}}\left(T - \frac{c}{N}\sum_{t=1}^{T}\frac{1}{t}\right) = C_{\text{err}}(T - O(\log T)).$$

The dominant part of this expression is $C_{\text{err}} \cdot T$. This means the regret incurred from making systematic errors during exploitation grows linearly with time. Combining both terms, the total expected regret is bounded by $\mathbb{E}[\mathcal{R}_T] \leq O(\log T) + O(T)$, which is ultimately dominated by the linear term, resulting in $\mathbb{E}[\mathcal{R}_T] \sim O(T)$.

### A.3 DQN Algorithm

The algorithm employs several technical components to enhance numerical stability and performance:

**Huber Loss (SmoothL1Loss):** Instead of using standard mean squared error (MSE), the algorithm employs Huber loss, which behaves like MSE for small errors but like mean absolute error (MAE) for large errors. Mathematically, it is defined as:

$$L_\delta(y, \hat{y}) = \begin{cases} \frac{1}{2}(y - \hat{y})^2 & \text{for } |y - \hat{y}| \leq \delta \\ \delta \cdot (|y - \hat{y}| - \frac{1}{2}\delta) & \text{otherwise} \end{cases} \tag{A.5}$$

where $\delta$ is a threshold parameter. This loss function is more robust to outliers in the TD error, which can be significant in the early stages of learning or when exploring new regions of state space. In PyTorch, this is implemented as *nn.SmoothL1Loss()*.

**Experience Replay:** The algorithm uses a replay memory to store transitions and randomly samples from this memory for training. This breaks the correlation between consecutive transitions and stabilizes learning. The replay memory size here is typically set to 20,000 transitions.

**Soft Target Network Updates:** Rather than periodically copying the policy network weights to the target network, the algorithm uses soft updates: $\theta^- \leftarrow \tau\theta + (1 - \tau)\theta^-$ with $\tau = 0.05$. This creates a more stable learning target.

**Gradient Clipping:** To prevent exploding gradients, the algorithm employs gradient clipping, restricting gradient values to a maximum magnitude of 100: *torch.nn.utils.clip_grad_value_(policy_net.parameters(), 100)*.

### A.4 Clipped Surrogate Objective Function

The clipped surrogate objective is the core innovation of the PPO algorithm, designed to address a fundamental challenge in policy gradient methods: controlling the step size of policy updates. Traditional policy gradient approaches can suffer from instability when policy updates are too large, potentially leading to sudden performance degradation and difficulty recovering to good policies.

**Mathematical Foundation**

The foundation of PPO lies in the importance sampling ratio, which measures how the probability of selecting the same action changes between the old and new policies:

$$r_t(\theta) = \frac{\pi_\theta(a_t|S_t)}{\pi_{\theta_{\text{old}}}(a_t|S_t)}$$

In standard policy gradient methods, the objective function is formulated as:

$$\mathcal{L}^{PG}(\theta) := \mathbb{E}_t[r_t(\theta)A_t],$$

where $A_t$ represents the advantage function. However, this formulation can lead to excessively large policy updates when the importance sampling ratio becomes too large or too small.

PPO addresses this issue through its clipped objective function:

$$\mathcal{L}^{\text{CLIP}}(\theta) = \mathbb{E}_t[\min(r_t(\theta)A_t, \text{clip}(r_t(\theta), 1 - \epsilon, 1 + \epsilon)A_t)].$$

**Clipping Mechanism**

The clipping mechanism operates based on the advantage function value and the importance sampling ratio:

$$\text{Clipped term} = \begin{cases} \min(r_t(\theta), 1 + \epsilon)A_t & \text{if } A_t > 0 \text{ (beneficial action)} \\ \max(r_t(\theta), 1 - \epsilon)A_t & \text{if } A_t < 0 \text{ (detrimental action)} \end{cases}$$

When $A_t > 0$, the clipping prevents the new policy from becoming overly biased toward this beneficial action by limiting the importance ratio to at most $1 + \epsilon$. When $A_t < 0$, clipping ensures the new policy does not excessively avoid this detrimental action by bounding the ratio to at least $1 - \epsilon$.

This design creates a conservative update mechanism where the objective function only improves when the policy change is within the trusted region defined by $[1 - \epsilon, 1 + \epsilon]$. When the importance sampling ratio falls outside this range, the gradient contribution is either clipped or set to zero, effectively preventing destructive policy updates.

**Standard Implementation**

The clipped surrogate objective is not a project-specific modification but rather the standard mechanism in PPO implementations. Most practical PPO applications use this approach with typical values of $\epsilon$ ranging from 0.1 to 0.3, with $\epsilon = 0.2$ being the most common choice. This universality stems from the method's simplicity, computational efficiency, and demonstrated stability across diverse reinforcement learning tasks.

A.5 FHN SYSTEM

In the FitzHugh-Nagumo model oscillation, the limit cycle naturally divides into two distinct phases, each characterized by a fast-slow dynamical structure. During the *recovery phase*, the trajectory drifts slowly down the left branch of the $x$-nullcline until it reaches the knee at approximately $x = -1$, where the $x$ variable undergoes a rapid jump to the right branch. In the *excitation phase*, the $x$ variable first jumps quickly toward the right branch and then moves slowly upward along that branch until reaching the upper knee near $x = 1$. This fast-slow alternation emerges because the $x$ variable evolves on an $\mathcal{O}(1)$ timescale as the fast subsystem while the $y$ variable evolves on an $\mathcal{O}(\varepsilon^{-1})$ timescale as the slow subsystem. The resulting dynamics create a characteristic oscillation pattern where each phase contains a rapid transition followed by gradual evolution along the corresponding nullcline branch, with phase transitions occurring at the turning points of the cubic nullcline where the trajectory direction fundamentally shifts between recovery and excitation processes.

A.6 MORE EXPERIMENTAL RESULTS OF FHN

Here we increased the noise parameters to $\sigma_1 = \sigma_2 = 0.5$, as shown in Figure 7. In this case, the limit cycle behaviour is not clear due to the large perturbation from noise; however, the unstable fixed point would be clearer than the small noise case.

A.7 DETAIL OF ASSUMPTION 5.5

This section details the reformulation of the Bellman equation for $V^\pi$ under the linear value function approximation from Assumption 5.7, framed in terms of linear operators. The Bellman equation for a policy $\pi$ is:

$$V^\pi(s) = \mathbb{E}_{a \sim \pi, s' \sim P}[R(s, a) + \gamma V^\pi(s')|s].$$

By substituting the linear forms $V^\pi(s) = w_\pi^T \mathbf{g}(s)$ and $\mathbb{E}[R(s, a)|s, \pi] = r_\pi^T \mathbf{g}(s)$, and using the linearity of expectation:

$$w_\pi^T \mathbf{g}(s) = r_\pi^T \mathbf{g}(s) + \gamma \mathbb{E}[w_\pi^T \mathbf{g}(s')|s, \pi].$$

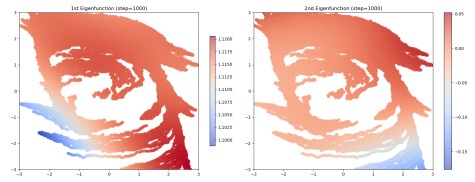 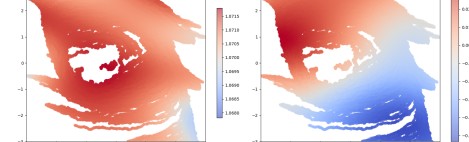

(a) 1st&2nd eigenfunctions at step 1000      (b) 1st&2nd eigenfunctions at step 4000

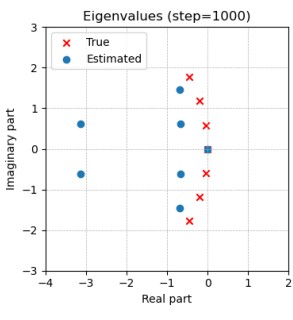 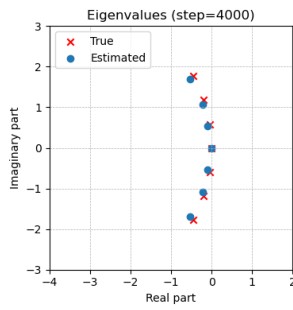

(c) Eigenvalues step at 1000      (d) Eigenvalues step at step 4000

Figure 7: 1st and 2nd eigenfunctions and eigenvalues of Koopman generator with noise $\sigma_1 = \sigma_2 = 0.5$.

The expectation term can be rewritten using the definition of the Koopman operator $\mathcal{K}_\pi$ and the inner product $\langle \cdot, \cdot \rangle$ in the feature space $\mathbb{R}^N$:

$$
\begin{aligned}
\mathbb{E}[w_\pi^T \mathbf{g}(s')|s, \pi] &= \langle w_\pi, \mathbb{E}[\mathbf{g}(s')|s, \pi]\rangle \\
&= \langle w_\pi, (\mathcal{K}_\pi \mathbf{g})(s)\rangle \\
&= \langle \mathcal{K}_\pi^* w_\pi, \mathbf{g}(s)\rangle = (\mathcal{K}_\pi^* w_\pi)^T \mathbf{g}(s),
\end{aligned}
$$

where $\mathcal{K}_\pi^*$ is the adjoint operator of $\mathcal{K}_\pi$. Substituting this back, we have:

$$
w_\pi^T \mathbf{g}(s) = r_\pi^T \mathbf{g}(s) + \gamma(\mathcal{K}_\pi^* w_\pi)^T \mathbf{g}(s).
$$

Since this must hold for all $s$, we can equate the vector coefficients:

$$
w_\pi = r_\pi + \gamma \mathcal{K}_\pi^* w_\pi \quad \implies \quad (I - \gamma \mathcal{K}_\pi^*) w_\pi = r_\pi.
$$

The uniqueness of the solution $w_\pi = (I - \gamma \mathcal{K}_\pi^*)^{-1} r_\pi$ is guaranteed by the invertibility of the operator $(I - \gamma \mathcal{K}_\pi^*)$. This requires the spectral radius $\rho(\gamma \mathcal{K}_\pi^*) < 1$.

As an expectation operator, $\mathcal{K}_\pi$ is a non-expansive mapping in the $L^\infty$-norm, implying its spectral radius $\rho(\mathcal{K}_\pi) \leq 1$. The adjoint operator has the same spectral radius, $\rho(\mathcal{K}_\pi^*) = \rho(\mathcal{K}_\pi)$. Therefore:

$$
\rho(\gamma \mathcal{K}_\pi^*) = \gamma \rho(\mathcal{K}_\pi^*) \leq \gamma.
$$

Since $\gamma \in [0, 1)$, we have $\rho(\gamma \mathcal{K}_\pi^*) < 1$, which ensures that the inverse operator $(I - \gamma \mathcal{K}_\pi^*)^{-1}$ exists and is bounded. This guarantees a unique solution for $w_\pi$.

# Appendix B

## B.1 PROOF OF THEOREM 5.3

*Proof.* The total expected regret is the sum of expected regrets from *exploration* and *exploitation* over $T$ steps: $\mathbb{E}[\mathcal{R}_T] = \sum_{t=1}^T (\epsilon_t \cdot \mathbb{E}[\Delta_{a_t}|\text{exploration}] + (1 - \epsilon_t) \cdot \mathbb{E}[\Delta_{a_t}|\text{exploitation}])$.

First, we analyze the *exploitation* term. A suboptimal arm $a'$ is selected during exploitation only if $\hat{R}_{a'} > \hat{R}_{a^*}$. This is only possible if the true gap satisfies $\Delta_{a'} < 2\varepsilon_{\text{sdmd}}$. This condition arises

because for a mistake to occur, the most optimistic estimate for the suboptimal arm $(R_{a'} + \varepsilon_{\text{sdmd}})$ must be greater than the most pessimistic estimate for the optimal arm $(R_{a^*} - \varepsilon_{\text{sdmd}})$.

However, Eq. equation 5.1 of Assumption 5.2 states that for all suboptimal arms $a'$, $\Delta_{a'} \geq \Delta_{\min} \geq 2\varepsilon_{\text{sdmd}}$. This creates a contradiction. Therefore, under Assumption 5.2, the condition $\Delta_{a'} < 2\varepsilon_{\text{sdmd}}$ is never met for any suboptimal arm. This means that during any exploitation step, the algorithm will never select a suboptimal arm. The expected regret from exploitation is thus zero:

$$\mathbb{E}[\Delta_{a_t}|\text{exploitation}] = 0.$$

The cumulative regret is therefore bounded solely by the cost of exploration:

$$\mathbb{E}[\mathcal{R}_T] \leq \sum_{t=1}^{T} \epsilon_t \cdot \mathbb{E}[\Delta_{a_t}|\text{exploration}] = \sum_{t=1}^{T} \epsilon_t \left( \frac{1}{N} \sum_{a=1}^{N} \Delta_a \right).$$

Using the decaying exploration parameter $\epsilon_t = c/(Nt)$, the bound becomes:

$$\mathbb{E}[\mathcal{R}_T] \leq \sum_{t=1}^{T} \frac{c}{Nt} \left( \frac{1}{N} \sum_a \Delta_a \right) = \frac{c}{N^2} \left( \sum_a \Delta_a \right) \sum_{t=1}^{T} \frac{1}{t}.$$

where the summation $\sum_{t=1}^{T} \frac{1}{t} \sim O(\log T)$. Thus, the expected cumulative regret is bounded by $O(\log T)$. As $\lim_{T\to\infty}(\log T)/T = 0$, the expected average regret converges to zero. $\qquad\square$

## B.2   PROOF OF LEMMA B.1

**Lemma B.1** (Bellman Target Estimation Error). Let $\mathcal{T}$ be the ideal Bellman optimality operator and $\hat{\mathcal{T}}$ be its empirical counterpart estimated by DQN-SDMD. Under Assumptions 5.1 and 5.5, and a uniform bound $\|Q_k\|_\infty \leq Q_{\max}$, the single-step estimation error is bounded as following:

$$\|(\mathcal{T}Q_k) - (\hat{\mathcal{T}}Q_k)\|_\infty \leq \gamma Q_{\max} \cdot \varepsilon_{\text{sdmd}} \quad \text{for all } k \geq 0. \tag{B.1}$$

*Proof.* The sup-norm is taken over all state-action pairs $(s, a)$, i.e., $\|f\|_\infty = \sup_{s,a} |f(s,a)|$. The error between the ideal and estimated Bellman targets arises solely from the approximation of the expectation term. Let us define the function $f_k(s) := \max_{a'} Q_k(s, a')$. The error is then:

$$\|(\mathcal{T}Q_k) - (\hat{\mathcal{T}}Q_k)\|_\infty = \sup_{s,a} \left| (R(s,a) + \gamma\mathbb{E}[f_k(s')|s,a]) - \left( R(s,a) + \gamma\widehat{\mathbb{E}}[f_k(s')|s,a] \right) \right|$$

$$= \gamma \sup_{s,a} \left| \mathbb{E}[f_k(s')|s,a] - \widehat{\mathbb{E}}[f_k(s')|s,a] \right|$$

$$= \gamma \sup_{s,a} \left| ((\mathcal{K}_a - \hat{\mathcal{K}}_a)f_k)(s) \right|$$

$$\leq \gamma \sup_{a} \|\mathcal{K}_a - \hat{\mathcal{K}}_a\| \cdot \|f_k\|_\infty.$$

Here, $\mathbb{E}[\cdot]$ is the true conditional expectation, estimated by SDMD as $\widehat{\mathbb{E}}[\cdot]$. The third line follows by representing these expectations via the action-conditioned Koopman operator $\mathcal{K}_a$ and its estimate $\hat{\mathcal{K}}_a$.

From our core Assumption 5.1, we have $\|\mathcal{K}_a - \hat{\mathcal{K}}_a\| \leq \varepsilon_{\text{sdmd}}$ for all actions $a$. The sup-norm of the function $f_k$ is $\|f_k\|_\infty = \sup_s |\sup_{a'} Q_k(s,a')| = \|Q_k\|_\infty \leq Q_{\max}$. Substituting these bounds into our inequality gives the final result. $\qquad\square$

## B.3   PROOF OF THEOREM 5.6

The proof proceeds in two parts:

*Part 1: Convergence of the Q-function.* This is a classic result from the analysis of Approximate Value Iteration (see Bertsekas (2012)). Let $\delta_k = \|Q^* - Q_k\|_\infty$. The error propagation for one step of the Q-learning update, where $Q_{k+1} \approx \hat{\mathcal{T}}Q_k$, is:

$$\|Q^* - Q_{k+1}\|_\infty = \|\mathcal{T}Q^* - \hat{\mathcal{T}}Q_k\|_\infty$$

$$\leq \|\mathcal{T}Q^* - \mathcal{T}Q_k\|_\infty + \|\mathcal{T}Q_k - \hat{\mathcal{T}}Q_k\|_\infty$$

$$\leq \gamma\|Q^* - Q_k\|_\infty + \tau_{Q,k}.$$

The first inequality is the triangle inequality. The second inequality uses the fact that the Bellman operator $\mathcal{T}$ is a $\gamma$-contraction in the sup-norm. Letting $\tau_Q = \sup_k \tau_{Q,k} = \gamma Q_{\max} \varepsilon_{\text{sdmd}}$, we have the recursion $\delta_{k+1} \leq \gamma \delta_k + \tau_Q$. As $k \to \infty$, this recursion converges to a fixed point satisfying $\delta_\infty \leq \gamma \delta_\infty + \tau_Q$, which implies $\delta_\infty \leq \frac{\tau_Q}{1-\gamma} \leq \frac{\gamma Q_{\max} \varepsilon_{\text{sdmd}}}{1-\gamma}$.

*Part 2: Bounding the Policy Suboptimal Gap.* To connect the Q-function error to the policy's performance gap, we use a standard result often called the Simulation Lemma (see Kearns & Singh (2002)). It states that for any policy $\pi$ that is greedy with respect to a Q-function $Q$, its value $J(\pi)$ is related to the optimal value $J(\pi^*)$ by:

$$J(\pi^*) - J(\pi) \leq \frac{2}{1-\gamma} \|Q^* - Q\|_\infty.$$

Applying this result to our case, where $\pi_k$ is the greedy policy for $Q_k$, we have:

$$J(\pi^*) - J(\pi_k) \leq \frac{2}{1-\gamma} \|Q^* - Q_k\|_\infty.$$

Taking the $\lim\sup$ as $k \to \infty$ and substituting the bound for $\|Q^* - Q_k\|_\infty$ from Part 1, we get:

$$\limsup_{k\to\infty} (J(\pi^*) - J(\pi_k)) \leq \frac{2}{1-\gamma} \left( \limsup_{k\to\infty} \|Q^* - Q_k\|_\infty \right)$$

$$\leq \frac{2}{1-\gamma} \left( \frac{\tau_Q}{1-\gamma} \right)$$

$$\leq \frac{2\gamma Q_{\max} \varepsilon_{\text{sdmd}}}{(1-\gamma)^2}.$$

## B.4 PROOF OF LEMMA B.2

**Lemma B.2** (Value Estimation Error Bound). Let $V^\pi$ be the true value function and $\hat{V}^\pi$ be its estimation computed via SDMD. Under Assumptions 5.1 and 5.7, the error is bounded as following:

$$\|\hat{V}^\pi - V^\pi\|_\infty \leq \tau, \quad \text{where } \tau = \mathcal{O}\left( \frac{\varepsilon_{\text{sdmd}}}{1-\gamma} \right). \tag{B.2}$$

*Proof.* We first bound the error in the weight vector, $\|\hat{w}_\pi - w_\pi\|$. The weights are defined by $w_\pi = (I - \gamma \mathcal{K}_\pi^*)^{-1} r_\pi$ and $\hat{w}_\pi = (I - \gamma \hat{\mathcal{K}}_\pi^*)^{-1} r_\pi$. Using the operator identity $(A^{-1} - B^{-1}) = A^{-1}(B - A)B^{-1}$, we have:

$$\hat{w}_\pi - w_\pi = \left[ (I - \gamma \hat{\mathcal{K}}_\pi^*)^{-1} - (I - \gamma \mathcal{K}_\pi^*)^{-1} \right] r_\pi$$

$$= (I - \gamma \hat{\mathcal{K}}_\pi^*)^{-1} \left[ (I - \gamma \mathcal{K}_\pi^*) - (I - \gamma \hat{\mathcal{K}}_\pi^*) \right] (I - \gamma \mathcal{K}_\pi^*)^{-1} r_\pi$$

$$= \gamma (I - \gamma \hat{\mathcal{K}}_\pi^*)^{-1} (\hat{\mathcal{K}}_\pi^* - \mathcal{K}_\pi^*)(I - \gamma \mathcal{K}_\pi^*)^{-1} r_\pi$$

$$= \gamma (I - \gamma \hat{\mathcal{K}}_\pi^*)^{-1} (\hat{\mathcal{K}}_\pi^* - \mathcal{K}_\pi^*) w_\pi.$$

Taking the vector norm on both sides and applying the properties of induced operator norms:

$$\|\hat{w}_\pi - w_\pi\| \leq \gamma \|(I - \gamma \hat{\mathcal{K}}_\pi^*)^{-1}\| \cdot \|\hat{\mathcal{K}}_\pi^* - \mathcal{K}_\pi^*\| \cdot \|w_\pi\|$$

$$\leq \gamma \frac{1}{1-\gamma} \cdot \varepsilon_{\text{sdmd}} \cdot \|w_\pi\|,$$

where we used the fact that for any Koopman operator $\mathcal{K}$, its spectral radius $\rho(\gamma \mathcal{K}) \leq \gamma$, which ensures $\|(I - \gamma \mathcal{K}^*)^{-1}\| \leq 1/(1-\gamma)$. Also, $\|\hat{\mathcal{K}}_\pi^* - \mathcal{K}_\pi^*\| = \|\hat{\mathcal{K}}_\pi - \mathcal{K}_\pi\| \leq \varepsilon_{\text{sdmd}}$ from Assumption 5.1. The value function error is then bounded by:

$$\|\hat{V}^\pi - V^\pi\|_\infty = \sup_{s\in\mathcal{S}} |(\hat{w}_\pi - w_\pi)^T \mathbf{g}(s)|$$

$$\leq \|\hat{w}_\pi - w_\pi\| \cdot \sup_{s\in\mathcal{S}} \|\mathbf{g}(s)\|$$

$$\leq \left( \gamma \frac{1}{1-\gamma} \varepsilon_{\text{sdmd}} \|w_\pi\| \right) \cdot C_{\mathbf{g}} = \tau.$$

where $C_{\mathbf{g}} = \sup_{s\in\mathcal{S}} \|\mathbf{g}(s)\|$. $\square$

## B.5 PROOF OF THEOREM 5.8

*Proof.* The proof applies the standard convergence theorem for Approximate Policy Iteration (API) Bertsekas & Tsitsiklis (2008); Bertsekas (2012). The API theorem states that if the policy evaluation step incurs a maximum-norm error of $\varepsilon$ (i.e., $\|\hat{V}^k - V^k\|_\infty \leq \varepsilon$), the asymptotic suboptimal gap is bounded by $\frac{2\gamma\varepsilon}{(1-\gamma)^2}$.

In our framework, the policy evaluation error $\varepsilon$ is not an abstract quantity but is determined by the SDMD procedure. According to Lemma B.2, this error is bounded by:

$$\varepsilon = \|\hat{V}^k - V^k\|_\infty \leq \tau = \frac{\gamma C_\mathbf{g} W_{\max}}{1-\gamma}\varepsilon_{\text{sdmd}}.$$

Substituting this specific error bound for $\varepsilon$ into the general API convergence result yields our final bound:

$$\limsup_{k\to\infty} \left(J(\pi^*) - J(\pi_k)\right) \leq \frac{2\gamma}{(1-\gamma)^2} \cdot \left(\frac{\gamma C_\mathbf{g} W_{\max}}{1-\gamma}\varepsilon_{\text{sdmd}}\right)$$
$$= \frac{2\gamma^2 C_\mathbf{g} W_{\max}}{(1-\gamma)^3}\varepsilon_{\text{sdmd}}.$$

This directly links the final policy's performance to the estimation accuracy of the SDMD algorithm. $\square$

