# OpenReview forum: "Reinforced Data-Driven Estimation for Spectral Properties of Koopman Semigroup in Stochastic Dynamical Systems"
_ICLR.cc/2026/Conference — ICLR 2026 Conference Withdrawn Submission_

### Official Review · Reviewer_hWgD · 2025-10-21

**Soundness:** 3
**Presentation:** 3
**Contribution:** 3
**Rating:** 6
**Confidence:** 3

**Summary:**

This paper introduces a novel technique for learning the Koopman operator via reinforcement learning (RL). The proposed framework, called Reinforced Stochastic Dynamic Mode Decomposition (Reinforced SDMD), integrates RL with SDMD to actively guide the data acquisition process in stochastic dynamical systems. The method leverages three RL algorithms (Multi-Armed Bandit, Deep Q-Network (DQN), and Proximal Policy Optimization (PPO)) to generate well-behaved trajectories that enhance the robustness of Koopman operator estimation. The reward signal is based on a spectral consistency criterion, designed to encourage the agent to collect informative trajectories while maintaining adequate exploration. The authors validate their approach on three synthetic systems (the double-well potential, stochastic Duffing oscillator, and FitzHugh–Nagumo model), showing that the agent can identify dynamically relevant regions. They also provide a theoretical convergence analysis that connects estimation accuracy to the quality of the learned sampling policy.

**Strengths:**

The integration of RL and Koopman operator estimation is conceptually appealing and addresses an important limitation of existing data-driven spectral methods, i.e., their dependence on data quality and sampling. I appreciate that this approach is systematically evaluated using three distinct RL algorithms and tested across three representative dynamical systems. Overall, the paper is clearly written, well structured, and technically sound.

**Weaknesses:**

- While the qualitative illustrations are convincing, it remains unclear how much improvement RL sampling yields compared to random or uniform sampling strategies. Including quantitative metrics, for instance the eigenvalue distance between estimated and ground-truth spectra (e.g., using the directed Hausdorff distance) would significantly strengthen the empirical validation.

- The experiments focus exclusively on 2D systems; even a moderate increase in dimensionality (e.g., 5-10 dims) would help demonstrate the method’s scalability and computational feasibility.

**Questions:**

See Weaknesses

---

> ### Author Response · Authors · 2025-11-26
>
> >1. While the qualitative illustrations are convincing, it remains unclear how much improvement RL sampling yields compared to random or uniform sampling strategies. Including quantitative metrics, for instance the eigenvalue distance between estimated and ground-truth spectra (e.g., using the directed Hausdorff distance) would significantly strengthen the empirical validation.
>
> We appreciate the reviewer’s suggestion regarding quantitative comparison with random or uniform sampling. These alternative sampling strategies are indeed interesting, and the idea of comparing reinforcement style sampling with a broader family of sampling algorithms could form a meaningful study on its own. However, such comparisons would constitute a separate research direction, as the primary goal of the present work is to introduce the first RL-style framework specifically aimed at learning Koopman spectral objects in stochastic systems, rather than benchmarking sampling approaches. At the same time, part of this comparison is indirectly reflected in our current results: traditional SDMD with any trajectory data often produces unstable spectra, whereas Reinforced SDMD computes better eigenvalue aligned with expected generator behavior after some iterations, which is precisely the qualitative improvement needed for spectral estimation. Given that many stochastic systems lack closed-form or analytical spectra, we focused on spectral structure and stability rather than direct numerical distances. We will clarify this motivation and also note that a wider and broader quantitative comparison of sampling strategies will be pursued as future work, since it becomes a new and independent topic beyond the scope of this paper.
>
> >The experiments focus exclusively on 2D systems; even a moderate increase in dimensionality (e.g., 5-10 dims) would help demonstrate the method’s scalability and computational feasibility.
>
> We agree with the reviewer that higher-dimensional experiments are important, and we appreciate this observation. Extending Reinforced SDMD to 5–10 or even higher dimensional systems is a nontrivial but meaningful in the next step, and we intend to explore this direction as future work. In the present submission, our primary objective is to establish a new RL-based perspective and methodology for Koopman spectral learning, which already introduces several conceptual novelties that required substantial development: neural estimation of drift–diffusion, automatic-differentiation-based generator construction, and iterative reinforced style updates. There do exist high dimensional Koopman learning studies that can be incorporated into our proposed Reinforced framework, but integrating those approaches with RL-driven adaptive sampling is itself a significant research project requiring extensive analysis and computational design. We want to indicate that high-dimensional extensions are valuable and well planned for future work, while the current paper focuses on demonstrating the conceptual and methodological contribution of the proposed RL-based Koopman spectral learning framework.

---

### Official Review · Reviewer_YQRw · 2025-10-24

**Soundness:** 2
**Presentation:** 2
**Contribution:** 2
**Rating:** 2
**Confidence:** 2

**Summary:**

The paper proposes Reinforced Dynamic Mode Decomposition, which introduces Reinforcement Learning (RL) to automatically guide the data collection process for Stochastic Dynamic Mode Decomposition. They shape a new reward signal to guide the agent based on the spectral consistency, which measures how well the Koopman operator has been estimated. The method has been validated using three different RL algorithms on canonical systems. Moreover, they provide a theoretical analysis of the proposed algorithm under strong assumptions.

**Strengths:**

- They introduce a new reward function to guide the agent in collecting the data. The function leverages an exploitation term, which is defined as the spectral consistency, and an exploration bonus measured with the Gaussian kernel.
- The data-collection method is shown to be working using three different RL algorithms.

**Weaknesses:**

1. Lack of baselines: The paper is missing baselines to show that their method brings benefits. For example, randomly initializing the agent position at each rollout and collecting data from there could be a simple yet effective comparison.
2. Inconsistency between the theory and the experimental results: The theoretical results are built upon the strong assumptions that the Q and V functions could be linearly expressed, respectively, in the DQN and PPO algorithms. However, these assumptions are not tested to see if they hold in practice.
3. Missing reward learning results: The paper does not demonstrate whether the proposed reward function can actually be learned by the agent in practice. The authors should include figures showing the results compared to the agent’s performance relative to the reward convergence, to illustrate how expert the used agent is for data collection.
4. Computational costs: The proposed method appears to be computationally intensive. Without a comparison to baseline methods, it remains unclear whether the computational costs are justified.
5. Unclear role of R_0: You don’t justify the role of this baseline, and whether it is a hyperparameter to be tuned or not. You left over this term along the paper.

Minor
1. In Line 356, you mention that Figure 5 shows the first eigenfunction, but you are showing just the second one.
2. Typo in line 483 -> “Essentiall”

**Questions:**

Unclear meaning of images and doubts on the evaluation process:
1. You show in Figures 2, 3, and 5 that the Koopman eigenfunctions are learnt better on the data points collected using an improved agent over training. Are these eigenfunctions learnt using a fixed number of points? Are these coming from the different policies obtained along the training in an “off-policy” way?
2. Where do the points on which you evaluate the eigenfunctions come from? Are the eigenfunctions learnt using all of that or just a portion of those points?

---

> ### Author Response · Authors · 2025-11-26
>
> >Lack of baselines:
>
> We acknowledge this concern. However, the primary contribution here is establishing the RL-based paradigm for Koopman spectrum learning, which is a fundamentally new approach not previously explored in the literature. Importantly, this is not a sampling method comparison but rather a novel methodology for Koopman spectral estimation, as there are no directly comparable RL-based Koopman learning method baselines in the Koopman operator literature. Our results demonstrate that RL agents successfully identify dynamically critical regions (equilibria, separatrices) without prior system knowledge.
>
> >Theory and experiment consistency
>
> The linear function approximation assumptions are standard theoretical tools in RL convergence analysis. They enable mathematical treatment and provide theoretical guarantees when such representations exist. Crucially, our overall convergence depends primarily on the SDMD component's convergence in large data and large dictionary functions, which has been rigorously established with probabilistic error bounds and finite-dimensional approximation guarantees [1,2]. As long as the Koopman operator estimation converges, which is theoretically guaranteed, the RL component guides the data collection toward improving this estimation, as our three empirical results demonstrate.
>
> [1] Korda, M., Mezić, I. On Convergence of Extended Dynamic Mode Decomposition to the [1] Koopman Operator. J Nonlinear Sci 28, 687–710 (2018).
>
> [2] Yuanchao Xu, Kaidi Shao, Isao Ishikawa, Yuka Hashimoto, Nikos Logothetis, Zhongwei Shen; A data-driven framework for Koopman semigroup estimation in stochastic dynamical systems. Chaos, (2025).
>
> >Missing reward learning results:
>
> We respectfully disagree with this characterization. It is important to clarify that DQN and PPO do not learn explicit reward maps over the entire state space (only Bandit does, as shown in Figure 1b), they learn policies for sequential decision-making. The success of our method is evaluated by the quality of the learned spectral properties (eigenvalues and eigenfunctions), which is precisely what most Figures have demonstrated through progressive improvement across training steps. This is fundamentally a spectral learning problem, not a sampling evaluation problem. The reward function serves as the training signal to guide the RL agent, and the learned spectral accuracy validates the effectiveness of this guidance.
>
> >Computational costs:
>
> We acknowledge that the computational intensity is heavy. However, this investment is justified in practical scenarios where traditional offline methods fail to capture the spectral structure accurately. Our framework enables continuous refinement: by iteratively minimizing spectral consistency through intelligent data collection, we can obtain high-quality spectral information even for systems where most non-RL based methods would require prohibitively large datasets or fail entirely. The computational cost trades off against the automation of expert knowledge and the ability to discover informative regions adaptively.
>
> >Role of $R_0$:
> We apologize for insufficient explanation. $R_0$ serves as a normalization constant to keep reward magnitudes numerically stable during training. It is not a sensitive hyperparameter which requirs careful tuning.
>
> >Q1\&Q2 About eigenfunction learning:
>
> The eigenfunctions are computed at each snapshot during training using the data collected up to that point under the current policy. At step 100, 500, 2000, etc., SDMD processes all accumulated trajectories to produce the eigenfunctions shown. In other words, the points in the figures represent all collected data up to that training iteration. Each RL action generates a trajectory, and these accumulate over time. The eigenfunction values are computed on this entire accumulated dataset, showing both where the agent has sampled and how well the approximation has been learned in those regions.
>
> Thanks for pointing out the issues in the minor part. I have fixed them. And yes, the 1st eigenfunction is just constant so I deleted the pictures while forgot to remove the statement. I have also deleted the statement accordingly.

---

> ### Comment · Reviewer_YQRw · 2025-11-27
> **Thank you for the answer!**
>
> Thank you for your clarification.
>
> > Missing reward learning results
>
> I agree on this point. Evaluating the agent's performance by showing the quality of the learnt operator is the right thing to do in this case.
>
> > Theory and experiment consistency
>
> I know that linearizing the Q and V functions is a standard theoretical tool in the RL literature to provide convergence analysis. However, in my experience, this assumption is difficult to hold in practice. Indeed, my claim is not on the theoretical analysis, but I would be more careful when correlating experiments and theoretical analysis, since the main theoretical assumption is broken in the experiments.
>
> > Q1&Q2 About eigenfunction learning
>
> It is still unclear if the figures are showing a fair comparison or not. It is clear that the agent samples trajectories closer to the "hard-to-learn" parts, but you are comparing the eigenfunctions learnt with different dataset sizes, which does not seem fair. Then,
>
> _The eigenfunctions are computed at each snapshot during training using the data collected up to that point under the current policy_
>
> It seems that the points do not come from a single ("current") policy, but these points are the aggregation of all the trajectories sampled, leveraging all the past agents. Sorry in advance if I am misinterpreting this point. Then, if I understand correctly, the advantage lies in learning the operator by accumulating trajectories coming from different agents rather than using trajectories sampled using a single trained agent.
>
> >Lack of baselines & Computational costs
>
> I think that these are the main issues of the proposed work. I understand that the goal is to propose a new RL approach for Koopman spectral learning and not claim superiority over other methods. However, a comparison is needed to help researchers understand if it is worth using your method or just to have an idea of performance. I still have this question: is it worth investing resources in computing an agent that tells where to start sampling trajectories, or should I randomly start?
>
> In the end, I thank the authors for engaging the discussion, but - at this point - I keep my score since the main issues were partially addressed.

---

### Official Review · Reviewer_yEup · 2025-10-31

**Soundness:** 3
**Presentation:** 1
**Contribution:** 2
**Rating:** 2
**Confidence:** 4

**Summary:**

The paper integrates Reinforcement Learning (RL) with Stochastic Dynamic Mode Decomposition (SDMD) to improve data-driven Koopman spectral estimation for stochastic dynamical systems. The method is named Reinforced Stochastic Dynamic Mode Decomposition (Reinforced SDMD).

It is well understood that the capacity of Koopman-based methods to approximate the spectral decomposition of the (possibly stochastic) evolution operator crucially depends on the distribution of samples, that is on where and how trajectories are obtained. Poorly chosen initial conditions or long-time scales to escape meta-stable regions lead to inaccurate estimation of eigenfunctions and spectra.

To address this issue, Reinforced SDMD casts data collection as an RL problem. That is, the agent’s policy determines the initial sampling locations of trajectories which are than obtained by numerically solving known SDE over some time-horizon.  The reward is based on spectral consistency, that is on how well estimated eigenpairs predict system evolution, and an exploration term encouraging coverage of the state space. The paper explores sequential decision-making algorithms including Bandit, DQN, and PPO, showing that the agent identifies dynamically informative regions.

Theoretical analysis that links the quality of the learned policy to Koopman operator estimation accuracy is provided. Experiments on small-dimensional canonical stochastic systems (double-well, Duffing oscillator, FitzHugh–Nagumo) show efficient discovery of coherent regions without prior domain knowledge.

**Strengths:**

(1) The proposal to use RL framework in combination with numerical solvers of SDEs to estimate the spectral decomposition of the corresponding semigroup of Markov transfer operators, and hence build an efficient ML based solver is, up to my knowledge, novel and interesting.

(2) The choice of analysing different sequential decision-making algorithms (Bandits, DQN, PPO) is appreciated.

(3) The choice of canonical SDEs is appropriate for small dimensional problems, and the experiments support the claim that  Reinforced-SDMD can obtain good estimation

**Weaknesses:**

(1) The paper misses to report on big body of work on learning transfer operators of stochastic systems related not only to proposed SDMD but also to the problem of sampling from complex distributions via data-driven SDEs. To name just a few directly related works:

- Christof Schütte and collaborators have a big track record on learning stochastic systems and in particular treating the problem this paper tackles, see e.g. 150p. review Overcoming the timescale barrier in molecular dynamics ActaNumerica 2023, and references therein.

- Frank Noe and collaborators have also made significant impact on this topic, see e.g. VAMPnets for deep learning of molecular kinetics, Nature Communications 2018.

- Massimiliano Pontil and collaborators made significant contributions in understanding learning algorithms of transfer operators associated to SDEs, see e.g. Learning dynamical systems via Koopman operator regression in reproducing kernel Hilbert spaces, NeurIPS2022.

- More recent papers providing methods on learning continuous semigroups of stochastic processes, also containing statistical learning bounds:
   - Hou et al., Sparse learning of dynamical systems in RKHS: An operator-theoretic approach. ICML2023
   - Devergne et al., From biased to unbiased dynamics: An infinitesimal generator approach. NeurIPS2024
   - Kostic et al., Laplace transform based low-complexity learning of continuous Markov semigroups. ICML2025

(2) Proposed method is not adequately compared to vanilla performance of numerical solvers of SDEs. Namely, the reader cannot judge if the overall computational complexity of Reinforced-SDMD that has a big overhead of using sequential decision-making algorithms to get new samples has any advantage of randomly sampling initial points in the space. This is even more so important, since the current paper only works in small state dimensions, and its scalability is not clear.

(3) Authors make strong Assumption on the core method of estimation of the transfer operator (SDMD), and then continue with standard theoretical arguments for convergence of  considered sequential decision-making algorithms. Hence, the main novelty of the paper is methodological with weak, at best, theoretical novelty.

(4) From my perspective, many aspects of the paper are not clarified enough, please see the questions for details.

(5) Given the of methods for learning representations with neural networks (VAMP-Nets, DPNets, LoRA,..) that are used as a subspace on which SDMD is run, as well as having many competitors of SDMD, at least broader discussion of different approaches that can be coupled with RL formulation is needed to appreciate the authors' proposed approach.

My current score reflects identified weaknesses, however I am ready to revise it depending on authors' clarifications and revision of the paper.

**Questions:**

(1) Your choice of reward implicitly assumes that the noise level (diffusion term) is much smaller than the signal of the drift, making the forecasting of states a reasonable task. But, isn't the forecasting of distributions an adequate reward for stochastic systems?  What happens if the diffusion is stronger, how it impacts the experiments?

(2) Looking at Eq. (2.7), one expects that the estimator has a large number of eigenvalues close to one when dictionary is sufficiently large. Is this the case in your experiments? If yes, how do you choose eigenvalue-eigenfucntion pairs which approximate true leading non-trivial ones (typically just few close to one).

(3) In Assumption 5.1, which norm is used? If the norm is operator norm in $\cal{F}=L^({\cal M},\rho)$, how you formally define estimator acting on this domain? If it is the norm w.r.t. finite-dimensional subspace of $\cal{F}$ given by the dictionary, transfer operator is typically not well-defined on that, so the assumption becomes unreasonable. I believe that formal presentation off the method needs to significantly improve in order that proofs

(4) For learning the transfer operator, or the generator, the samples need to come from the invariant distribution so that the one can guarantee learning of the object on the domain $\cal{F}$. However, in your proposal, we are getting samples from adequate supports of the invariant distribution (dense in meta-stable states), but I don;'t see how the samples are distributed according to $\rho$. What am I missing?

---

> ### Author Response · Authors · 2025-11-26
>
> Thank you for your thorough and insightful review. We greatly appreciate your detailed feedback and the opportunity to clarify our work. Below we address each of your concerns.
>
> >Missing Literature:
>
> We sincerely thank you for pointing out this significant gap in our literature review. You are absolutely correct that we missed important related works. We will incorporate proper citations as you suggested in the revised version. While these works focus on developing better estimators for transfer operators, our contribution addresses a complementary question: where to collect data to improve any given estimator's performance. Our framework is compatible to the underlying Koopman estimation method, which means that, one could replace SDMD with VAMPnets, etc. or other methods. We will clarify this positioning in revision later.
>
> >Comparison with Random Sampling:
>
> The goal of this initial work is not to claim computational superiority over random sampling but to show that RL policies identify dynamically informative regions (metastable states, transition pathways). In addition, scalability and large-scale performance are important future directions, but they are beyond the scope of this first methodological study.
>
> >Clarification on Assumption 5.1:
>
> Thank you for pointing this out. The norm $\mathcal{K} - \hat{\mathcal{K}}$ refers to the estimation error restricted to the finite-dimensional subspace $\mathcal{F}_N$ spanned by the dictionary. We have highlighted this part in the beginning of Section 5, which is in line397 right above Assumption 5.1. We do not assume the operator is bounded on the full $L^2$ space. Our theoretical contribution is that this creates a rigorous framework where improving data quality (via RL) directly translates to better spectral estimation, which is a connection not previously formalized in literature.
>
> >Comparison with Other Methods:
>
> We understand the desire for broader comparison with alternative estimators (VAMPnets, DPNets, LoRA, etc.) coupled with RL. Our contribution operates at the framework level rather than the algorithm level. The core scientific question we address is: "Can reinforcement learning automatically help Koopman operator to discover dynamical locations?"
>
> > the invariant distribution:
>
> This is a critical distinction in our framework. While the Koopman operator is theoretically defined on the space $L^2(\rho)$, the estimation of its infinitesimal generator $\mathcal{A}$ via SDMD relies on the local approximation of drift and diffusion dynamics using short-time trajectory bursts; this process requires sufficient samples at a given location to estimate the local derivatives, but it does not require the global dataset distribution to strictly converge to the invariant measure $\rho$. In fact, for example if we consider metastable systems, relying solely on samples from $\rho$ is often inefficient because transition regions (such as saddle points) have negligible probability mass and are thus not likely to be sampled. Our Reinforced-SDMD framework essentially works as an sampling strategy which can deliberately biases data collection towards these dynamically significant guided by the spectral error, which allows us to stitch together an accurate global approximation of the operator from local estimates more efficiently than passive sampling from $\rho$ case.
>
>
> >Questions:
>
> The norm in Assumption 5.1 is operator norm.

---

### Official Review · Reviewer_Bbwv · 2025-11-06

**Soundness:** 2
**Presentation:** 2
**Contribution:** 3
**Rating:** 2
**Confidence:** 5

**Summary:**

In the present paper, the authors introduce an amendment to the stochastic dynamic mode decomposition in which reinforcement learning is utilized in an outer loop to optimally select the trajectory sample points to satisfy a Koopman reward function. Evaluation is performed on a set of 3 stochastic dynamical system namely the double-well potential, the stochastic Duffing oscillator, and the FitzHugh-Nagumo model.

**Strengths:**

The paper is well-motivated, and very well written in its construction of the individual algorithm variants, relating the background theory for the Koopman operator to the subsequent construction of the algorithm variants.

**Weaknesses:**

As is the paper suffers from a number of inherent flaws, in short at a high level:
* Insufficient relation to existing work on optimal/intelligent sampling, especially also to active sampling into the broader purview of which this paper falls
* Lack of a coherent design of evaluation
* Approach seemingly confined to a single DMD algorithm. Broader utility to the field not readily apparent

#### Insufficient relation to related work
* The paper is not set in relation to other work which utilizes reinforcement learning for optimal sampling strategies such as Zhao and Pilai [1], and Shen et al. [2].
* The core idea of the paper, moving away from random sampling for sDMD, is one not only confined to DMD but the wider scientific computing and machine learning based design literature. Some use GP surrogates to sample from to alleviate expensive individual sampling trajectories [3], others train individual sampling models [4], yet all of them can be broadly summarized under the larger umbrella of _Active Learning_ [5]. The paper sadly fails to draw these connections.
* Line 54-56, the choice of dictionary is orthogonal to the same issue faced in symbolic regression / SINDy-based approaches. Drawing this connection would aid greatly in embedding this work in existing literature.
* Line 58, some have started utilizing LLMs for the learning of the dictionary. See e.g. [6].

#### Design of Evaluation
* The current evaluation does not permit to draw conclusions on the performance of the introduced algorithms as only the potential and eigenvalues of the stochastic dynamical systems are shown. Going further, it is not apparent which algorithm(s) are used for the construction of the eigenvalues.
* There exists no actual performance evaluation, such as e.g. evaluating each of the 3 algorithms on each of the stochastic dynamical system for its sampling efficiency to reach a predetermined quality. This would also be a natural point to introduce ablations.
* The authors perform no ablations. To properly motivate the use of reinforcement learning for optimal sampling, an ablation to random sampling and e.g. importance sampling would be highly desirable to actually establish the performance advantage of the introduced algorithms. As is, it is not apparent whether the new algorithms outperform a random sampling baseline or not.

#### References
1. Zhao, D., & Pillai, N.S. (2024). Policy Gradients for Optimal Parallel Tempering MCMC. ArXiv, abs/2409.01574.
2. Shen, W., Dong, J., & Huan, X. (2023). Variational Sequential Optimal Experimental Design using Reinforcement Learning. ArXiv, abs/2306.10430.
3. Jones, A., Cai, D., Li, D. et al. Optimizing the design of spatial genomic studies. Nat Commun 15, 4987 (2024). https://doi.org/10.1038/s41467-024-49174-4
4. Fannjiang C, Listgarten J. Autofocused oracles for model-based design. Advances in Neural Information Processing Systems. 2020;33:12945-56.
5. Hsu, D.J. (2010). Algorithms for active learning.
6. Grayeli, A., Sehgal, A., Costilla-Reyes, O., Cranmer, M.D., & Chaudhuri, S. (2024). Symbolic Regression with a Learned Concept Library. ArXiv, abs/2409.09359.

**Questions:**

* Why are the algorithms not evaluated on more challenging (stochastic) environments? As is, it is hard to evaluate the limits of the approach.
* Has there been any quantitative comparison to commonly used sampling algorithms? Training a reinforcement learning agent is not cheap, and especially in such a highly specific application it is not readily apparent to the reviewer how the expended compute is to be amortized later on. A potential comparison here might be taking one of your existing environments on which the agent is trained, and providing the same compute budget to a random sampling based SDMD to then compare them on key metrics.

---

> ### Author Response · Authors · 2025-11-25
>
> Thank you for providing the comments. I think that our work are mistakenly interpreted as some method for optimal sampling or active learning community. We would like to clarify that the central goal of the paper is not to design better sampling policies, but to introduce the first reinforcement-learning (RL) framework that directly learns the spectrum of the Koopman operator.
> Sampling occurs only implicitly as the Koopman semigroup evolves the trajectories; however, the RL agent’s objective is to minimize Koopman spectral error, not to optimize sampling efficiency. Thus, existing works in optimal sampling address fundamentally different goals.
>
> Although SDMD is used in the experiments, the RL formulation is fully general and applies to EDMD, gEDMD, ResDMD, or any Galerkin projection-based methods, because the RL loss depends only on spectral consistency, not the specific regression formula.
>
> However, the sampling-related papers you mentioned do appear relevant, and I will cite them in the revised related work section.
>
> **Response to Questions:**
> >Q1: Why are the algorithms not evaluated on more challenging (stochastic) environments?
>
> We chose double-well Duffing systems and FHN because they are canonical stochastic benchmarks with known spectral structure, that are widely used in Koopman generator studies (e.g., SDMD, gEDMD, JetEDMD, etc.). These systems are not easy to capture if one considers using neural network to learn the dictionary instead of manually selecting very special choices of dictionary (manually selecting dictionary would lack of interpretability and not practical). However, using RL strategy as shown in our work would address such issue.
>
> >Q2: Has there been any quantitative comparison to commonly used sampling algorithms?
>
> We appreciate this question as it allows us to clarify a fundamental aspect of our contribution.
> Our work represents the first paradigm for learning Koopman spectral properties through reinforcement learning. The appropriate comparison is not against general-purpose sampling algorithms, but rather against existing methods in the Koopman spectral estimation literature. In practice, standard neural network-based approaches (e.g., EDMD-DL, SDMD-DL) often struggle to produce accurate and interpretable eigenfunctions, particularly for systems with complex dynamical structures such as metastability or multi-scale dynamics. The learned eigenfunctions frequently suffer from: (1)Poor convergence or instability during training; (2) Sensitivity to initial conditions and hyperparameters; (3) Failure to capture the correct partition of state space. To our knowledge, no previous work or literature has attempted to use RL as a spectral-learning mechanism.

---

### Author Response · Authors · 2025-11-27
**Global comment**

We would like to respectfully clarify the intended scope of our work, as several comments seem to assume that the contribution lies in proposing a new sampling method. In fact, our main goal is to introduce the **first** RL-based framework specifically designed for **learning the Koopman operator and its spectral objects** in stochastic systems. The RL agent is driven by a reward that reflects spectral consistency, which allows the method to automatically identify **regions of the state space that contain the most informative dynamical structure** for Koopman analysis.

The sampling behavior that emerges from this process is therefore a **byproduct** of optimizing for Koopman spectral learning rather than an objective in itself. For this reason, comparisons of sampling efficiency or RL training curves fall outside the main focus of the paper. We clarified this perspective in the Remark 3.2 of revised manuscript, and we sincerely appreciate the reviewers’ feedback.

---

### Note · Authors · 2026-01-10

I have read and agree with the venue's withdrawal policy on behalf of myself and my co-authors.